# Cardiac Involvement in Classical Organic Acidurias: Clinical Profile and Outcome in a Pediatric Cohort

**DOI:** 10.3390/diagnostics13243674

**Published:** 2023-12-15

**Authors:** Silvia Passantino, Serena Chiellino, Francesca Girolami, Mattia Zampieri, Giovanni Battista Calabri, Gaia Spaziani, Elena Bennati, Giulio Porcedda, Elena Procopio, Iacopo Olivotto, Silvia Favilli

**Affiliations:** 1Department of Paediatric Cardiology, Meyer Children’s Hospital IRCCS, 50139 Florence, Italy; serenachiellino90@gmail.com (S.C.); francesca.girolami@meyer.it (F.G.); gb.calabri@meyer.it (G.B.C.); giaia.spaziani@meyer.it (G.S.); elena.bennati@meyer.it (E.B.); giulio.porcedda@meyer.it (G.P.); iacopo.olivotto@unifi.it (I.O.); silvia.favilli@meyer.it (S.F.); 2Inborn Metabolic and Muscular Disorders Unit, Meyer Children’s Hospital IRCCS, 50139 Florence, Italy; elena.procopio@meyer.it

**Keywords:** cardiomyopathy, organic acidemias, pediatrics

## Abstract

Background: Cardiac involvement is reported in a significant proportion of patients with classical organic acidurias (OAs), contributing to disability and premature death. Different cardiac phenotypes have been described, among which dilated cardiomyopathy (DCM) is predominant. Despite recent progress in diagnosis and treatment, the natural history of patients with OAs remains unresolved, specifically with regard to the impact of cardiac complications. We therefore performed a retrospective study to address this issue at our Referral Center for Pediatric Inherited Errors of Metabolism. Methods: Sixty patients with OAs (propionic (PA), methylmalonic (MMA) and isovaleric acidemias and maple syrup urine disease) diagnosed from 2000 to 2022 were systematically assessed at baseline and at follow-up. Results: Cardiac anomalies were found in 23/60 OA patients, all with PA or MMA, represented by DCM (17/23 patients) and/or acquired long QT syndrome (3/23 patients). The presence of DCM was associated with the worst prognosis. The rate of occurrence of major adverse cardiac events (MACEs) at 5 years was 55% in PA with cardiomyopathy; 35% in MMA with cardiomyopathy; and 23% in MMA without cardiomyopathy. Liver transplantation was performed in seven patients (12%), all with PA or MMA, due to worsening cardiac impairment, and led to the stabilization of metabolic status and cardiac function. Conclusions: Cardiac involvement was documented in about one third of children diagnosed with classical OAs, confined to PA and MMA, and was often associated with poor outcome in over 50%. Etiological diagnosis of OAs is essential in guiding management and risk stratification.

## 1. Introduction

Organic acidurias or organic acidemias (OAs) are a class of inherited metabolic disorders characterized by the accumulation of organic acids in body fluids. More than 65 organic acidurias have been described. The defect results from deficiency of a specific enzyme or of a transport protein in one of several cellular metabolic pathways involved in the catabolism of amino acids, carbohydrates or lipids [1,2].

The incidence of OAs varies in accordance with the study population based on age, referral patterns and ethnicity. The estimated incidence ranges are between 1:10,000 and 1:1,000,000 live births.

The most common inborn errors of branched-chain amino acid (BCAA) catabolism, are also defined as “classical OAs” and include propionic (PA), methylmalonic (MMA) and isovaleric (IVA) acidemias (or acidurias) and maple syrup urine disease (MSUD) [3]. These disorders are associated with a wide range of clinical manifestations, such as cardiomyopathy, liver diseases, pancreatitis, neutropenia, thrombocytopenia, hypotonia, lethargy, coma, ataxia, seizures, vomiting, developmental delay, osteomalacia and osteoporosis [4]. The age of onset and severity of clinical picture are highly variable.

Cardiac involvement, mostly represented by DCM, has been reported in several OAs and may negatively affect prognosis, eventually leading to premature death [5,6]. Among OAs, PA is the form which is most commonly associated with cardiac disease. Different types of cardiomyopathies (CMPs) have been described, namely dilated or hypertrophic; however, OAs may also manifest with arrhythmic disorders (acquired long QT syndrome) [7]. Cardiac dysfunction ranges from mild to progressive deterioration of left ventricular ejection fraction and chronic heart failure. The pathophysiology of cardiac complications has not been completely elucidated, but is thought to involve an interplay of mitochondrial dysfunction (bioenergetic deficiency and oxidative damage), miRNA deregulation, histone deacetylase (HDAC) and inhibition by propionate [8]. Despite improvements in therapeutic options for OAs, the overall outcome remains suboptimal due to cardiac, neurological and muscular complications, among others. The first objective of pharmacologic therapy, especially in patients with PA and MMA, is to prevent secondary carnitine deficiency by providing carnitine supplementation. The maintenance treatment includes L-carnitine supplementation, a low-protein diet and precursor-free synthetic amino acid mixtures, minerals and micronutrients. Despite the use of effective therapy with a low-protein diet and carnitine, the overall outcome is poor. Liver transplantation is recognized as a treatment option for patients with PA and those with MMA without renal impairment, but experience is limited, and the option is reserved for ultra-selected patients [9,10]. Because of the limited evidence available, natural history studies in PA patients are still urgently needed. Based on the vast experience of our tertiary metabolic disease referral center, we therefore performed the present study to assesses the prevalence and severity of cardiac involvement in a cohort of children with OAs followed over two decades.

## 2. Materials and Methods

We retrospectively analyzed clinical, biochemical and molecular records of 258 patients diagnosed with inborn errors of metabolism, referred to the Inherited Metabolic Disorders Unit at Meyer Children’s Hospital from 2000 to August 2022. From a population of 108 patients with OAs, 60 diagnosed with classical OAs were selected and represent the study cohort. Patients with MMA associated with homocystinuria were excluded because of different etiology (this disorder is caused by a congenital defect in intracellular cobalamin metabolism). 

The diagnostic pathway included clinical history, examination, laboratory work-up and genetic tests (Table 1). Family history was determined by investigating the occurrence of neonatal death/stillbirth and recurring features among other family members, with particular attention paid to cardiac, neurologic, renal and muscular manifestations, as well as establishing the transmission pattern [3,11]. 

In cases of suspected OAs based on clinical presentation, laboratory parameters or newborn screening, the diagnosis was confirmed by measurements of organic acids in body fluids. The gold standard was represented by urinary organic acid analysis, which also allowed us to define the specific subtype of organic acidemia. This analysis was usually performed on an overnight urine sample, but was also repeated during metabolic crises in selected patients. Confirmation of diagnosis was achieved by enzymatic activity assay or genetic testing. 

Diagnosis of MMA was confirmed by DNA sequencing of associated genes, with detection of carrier status in the parents. Isolated MMA can be caused by defects in four different genes, including MMUT, MMAA (cblA), MMAB (cblB) and MMADHC (cblD-MMA). cblA and cbLB variants are responsive to cobalamin supplementation. Many of the known MMUT mutations were functionally tested with specific assays, to differentiate mut° from mut− subtypes. 

The subtype mut0 is unresponsive to hydroxocobalamin supplementation in a propionate incorporation assay, while the mut− subtype is characterized by increased propionate incorporation after the addition of hydroxocobalamin to the cell culture medium. 

Diagnosis of PA was confirmed by molecular genetic tests and by detection of deficient PCC enzymatic activity or biallelic mutations in PCCA gene on chromosome 13q32.3 and PCCB gene on chromosome 3q22.3. There is no evidence of a genotype–phenotype correlation in PA, so the assessment of gene defect (in PCCA or PCCB) is not decisive in driving clinical management or prognosis.

In patients with IVA, an autosomal recessive disease, the biochemical diagnosis was generally confirmed by demonstrating mutations in the IVD gene on chromosome 15q15.

The most frequent missense mutation was p.A282V on the IVD gene. Diagnosis of IVA was confirmed by various direct and indirect functional tests to assess IVD activity, in combination with molecular genetic analyses. The diagnosis of MSUD was confirmed by identification of biallelic pathogenic variants in BCKDHA, BCKDHB or DBT [12]. Patients are usually homozygous or compound heterozygous for mutations in the same gene; there is no evidence of heterozygotes for mutations in two different genes.

Baseline clinical evaluation was defined as the time when the diagnosis of OAs was first confirmed. Demographic and clinical information was collected. All patients were assessed by echocardiography, 12-lead ECG and Holter ECG, within 6-12 months from the first clinical evaluation. Two-dimensional, Doppler and M-mode echocardiography were performed at rest according to standard protocols. On the 12-lead ECG, QTc intervals were measured manually from lead II (three consecutive cycles were measured to calculate a mean value). The QT interval was defined as the interval between the beginning of the QRS complex and the end of the T wave. Onset and offset of T wave were defined as the intersections of the isoelectric line and the tangent of the maximal slope on the up and down limbs of the T wave. QTc intervals were calculated according to Bazett’s formula. 

CMR examination was performed in selected patients with cardiac involvement, using commercially available scanners and workstations. Left ventricular end-diastolic and end-systolic volumes and left ventricular mass were obtained, and the presence of delayed enhancement was assessed by visual inspection 15 min after intravenous administration of gadolinium-based contrast agent. 

When cardiac involvement was present, laboratory tests were performed, including serum levels of pro B-type natriuretic peptide (NT-proBNP) and cardiac troponin I levels.

### 2.1. Definition of Cardiac Involvement

-Dilated cardiomyopathy (DCM) was diagnosed in the presence of increased ventricular end-diastolic diameter (>2 Z-Score) associated with any degree of systolic dysfunction, defined as impaired left ventricular ejection fraction (EF). EF was measured by the biplane Simpson method (normal value: >55%, mild dysfunction: 45–54%, moderate dysfunction: 30–44%, severe dysfunction: <30%).-Evidence of supraventricular or ventricular arrhythmias.-Otherwise unexplained Qtc prolongation (>460 ms).

Patients were followed up in a standard fashion at 6-month intervals by clinical examination, neurologic evaluation and metabolic assessment. Cardiologic follow-up was planned every 3–6 months in the presence of CI; otherwise, every 12 months. Major adverse cardiac events (MACEs) and death from all causes were assessed. MACEs included advanced heart failure and death related to heart failure (HF) and sudden cardiac death, defined as an unexpected collapse occurring in patients with a previously uneventful clinical course.

### 2.2. Statistical Analysis

Statistical analysis was performed using SPSS version 21 (IBM Corporation, Armonk, NY, USA) and R version 3.3.1 (R Foundation, Vienna, Austria). Data are expressed as percentage, mean and SD, or median with interquartile range (IQR) for skewed distributions. Multivariate regression models (Probit) were used to compare patient subsets with and without CI. All tests were two-tailed, and P-values less than 0.05 were considered significant. Kaplan–Meier curves were used to compare survival between the two groups (with and without CI) and to show MACE-free survival in CI group. STATA statistical software version 15 was used for all analyses.

## 3. Results

### 3.1. Baseline Characteristics

Among 60 patients with classic OAs, of whom 32 (53%) were males, 25 received a diagnosis of MMA, 20 of PA, 9 of IVA and 6 of MSUD. At diagnosis, the median age of the cohort was 36,432 days (IQ 0;15 years). Diagnosis was achieved by newborn screening (NBS) in 17 of the 60 patients (28%) (Figure 1). Overall, 33 of 60 patients (55%) had onset of OA in infancy or childhood, and only three patients (5%) developed symptoms during adolescence, around the age of 15. A family history of OAs was present in seven patients (12%).

### 3.2. Genetic Results

In 39 of 60 patients (65%) the diagnosis of OA was confirmed by molecular analysis, including 23 of 25 patients (92%) with MMA; the c.2194_2197delinsTGGAA variant was the most frequent variant (26%), followed by c.643G>A (22%). We performed molecular genetic testing in 12 of 20 patients with PA: the pathogenic variant (c.229C>T; p.Arg77Trp) in PCCA was the most common cause (25%), followed by the variant (c.337C>T; p.Arg113*) in PCCB (17%). 

We performed molecular genetic testing in seven of nine patients (78%) with IVA: the most frequent mutations on IVC were c.860G>A (29%) and c.941C>T (29%). Four of the six patients with MSUD were identified at neonatal screening, and in all of them, the genetic defect was confirmed by next generation sequencing (NGS).

### 3.3. Extra-Cardiac Involvement

Neurologic manifestations were the dominating feature, observed in 28 of the 60 patients (47%). 

The most recurrent neurologic disorders in our cohort were intellectual impairment and epilepsy; one patient presented West syndrome. Eleven of these twenty-eight patients had concomitant cardiac involvement. Ocular disorders were detected in five patients (8%), including myopia in two and optic atrophy in three. 

Hematologic abnormalities (anemia, thrombocytopenia, hypereosinophilia, thrombosis) were observed in five patients (8%). Gastrointestinal complaints such as feeding difficulties, gastrointestinal reflux and recurrent vomiting and pancreatitis were found in seven patients (12%). Failure to thrive and short stature were detected in four patients, associated with GH deficiency, early puberty and hypothyroidism. Other endocrinological changes included liver steatosis and fasting hyperglycemia. Skeletal abnormalities were observed in six patients (10%), including scoliosis, osteopenia, microcephaly, genu varus and pectus excavatum. Renal failure was detected in five patients (8%) of this cohort, and three of them received diagnoses of MMA. Two of these five patients, diagnosed with MMA and MSUD, presented acute renal failure after liver transplantation, and one with MMA required simultaneous liver–kidney transplantation (Table 2).

### 3.4. Cardiac Involvement

Cardiac involvement was present in 23 patients (38%) with a mean age at onset of cardiac manifestations of 10.3 years (range: 0.1–17.5), including 17 patients with PA and 6 with MMA; cardiomyopathies and acquired long QT were the most frequent findings, observed in 20 out of these 23 (86%). In addition, congenital heart diseases (atrial septal defect, patent ductus arteriosus and aortic bicuspid valve disease) were detected in three patients with MMA, but were considered associated lesions and not part of the OA spectrum. Apart from this, none of the patients with IVA and MSUD showed evidence of cardiac involvement. 

DCM was diagnosed by echocardiography in 20 patients at a median age of 10.3 years (range: 0–17.5); of these, 17 had PA. In all 20, the diagnosis was made by routine echocardiography, in the absence of overt heart failure, and was not associated with metabolic crises. Four patients showed pathological patterns of mitral valve inflow compatible with LV diastolic dysfunction.

ECG abnormalities were documented in 15 of the 20 patients with cardiomyopathy (75%). A prolonged QT interval associated with DCM was detected in 17/20 (85%); the mean QTc for the whole cohort was 465.5 ms (+/−7.2 ms SDS); 14 of 17 patients (82%) had at least one documented maximum QTc of ≥460 ms; and in 7 patients (41%), patients’ maximum QTc was ≥480 ms. None of the patients had documented ventricular arrhythmias or SCD.

Cardiac magnetic resonance was performed only in three adolescents: one had an abnormal ECG and cardiac symptoms, and the other two had DCM; none had late gadolinium enhancement (Figure 2).

### 3.5. Management

In all patients, medical treatment consisted of a low-protein diet, combined with a specific amino acid mixture as indicated. All received carnitine supplements (50 mg/kg/day). Cardioactive treatment was prescribed in 16 (80%) of the 20 patients with CMP, including 15 (75%) on beta-blockers (carvedilol or bisoprolol); 16 (80%) on ace-inhibitors (captopril, enalapril); and 7 (35%) on diuretics (furosemide and spironolactone). In six patients with progressive worsening of systolic function despite medical therapy, liver transplantation (LT) was performed. Only in one, with chronic renal failure but no cardiac involvement, a combined liver–kidney transplantation was performed. All transplanted patients were alive after a mean follow-up of 36 months (range: 1–48), with stabilization of their metabolic status; LVEF, however, improved slightly only in one patient, and did not change significantly in the remaining five. None had MACEs during follow-up.

### 3.6. Long-Term Outcome

Over a median observation period of 5 years, 11 out of the 20 patients with cardiac involvement presented a MACE (n = 5) or died due to HF (n = 2). Three patients died from non-cardiac causes (Table 2). The rate of occurrence of MACEs at 5 years, calculated using the Kaplan–Meier method corrected for left-truncation, was 55% in PA with CMP, 35% in MMA with CMP and 23% in MMA without CMP. The rate of MACEs in PA without cardiac involvement was low. Specifically, the annual rate of MACEs was 10.1% for PA with CMP, 7.2% for MMA with cardiomyopathy and 3.9% for MMA without CMP (Figure 3). Progressive systolic impairment with moderate or severe depression in LVEF (≤40%) was recognized in 9/20 patients, despite maximized pharmacological treatment. Notably, metabolic decompensation was associated with QTc prolongation, in five patients, but not with worsening of systolic function. Measurements of plasma NT-proBNP levels were normal in all but one patient with severe systolic dysfunction, in whom NT-proBNP was consistently high. 

## 4. Discussion

Cardiac involvement was documented in one third of patients with classical OAs consecutively diagnosed and treated at a referral center for pediatric IEM over 20 years. DCM was particularly common in patients with PA and—to lesser extent—with MMA, although the underlying mechanisms remains unclear, particularly as cardiac decompensation occurred independently of metabolic status. DCM with a low ejection fraction was a main driver of MACEs in our cohort and accounted for a considerable annual rate of MACEs during follow-up, as high as 10.1% for PA and 7.2% for MMA. Conversely, other etiologies such as IVA and MSUD showed virtually no propensity for cardiac involvement and were not associated with MACEs. This diversity suggests the possibility of differential follow-up strategies for patients with OAs, focusing appropriately on the etiologies at risk.

Acquired long QT was detected in most patients with PA, although younger patients appeared to be less prone, suggesting that some degree of cardiac involvement reflects the duration of disease as part of an ongoing progressive process. Worsening of QTc interval prolongation was more common during metabolic decompensation. However, no life-threatening arrhythmias were reported.

There is no strong evidence of a correlation between metabolic status or residual enzymatic activity and the onset of CMP [13]. In our experience, no relationship was found between metabolic decompensation and cardiac dysfunction. This suggests that other pathological mechanisms might be involved, including mitochondrial dysfunction, oxidative stress and changes in gene expression. Experimental studies have determined that systemic acid–base disturbances can profoundly affect the heart. Organic anions that accumulate in body fluids can collectively reach concentrations of several millimolar in severe metabolic acidosis, as in the case of inherited OAs, and exert powerful biological actions on the heart that are not well understood. Furthermore, the tricarboxylic acid cycle contributes to the normal contractility of the myocardium [14,15], and propionic acid represents a major anaplerotic substrate in the energetic metabolism of the heart, where PCC is expressed [16,17].

Medical treatment can stabilize cardiac function and should be started at the early stage of ventricular dysfunction [18]. The goals of cardiac therapy are to reduce the symptoms of HF and prevent disease progression, while at the same time improving the quality of life of symptomatic DCM patients [3,19].

Liver transplantation (LT) has been proposed as an effective therapeutic option, particularly in PA patients, because the liver is the main site of branched chain amino acid metabolism and consequently of propionic acid production [20,21]. However, the role of LT in PA is still controversial, mostly due to the presence of extrahepatic sources of propionic acid (i.e., the gastrointestinal tract and the central nervous system), leading to only partial correction of the metabolic defect. Indeed, LT helps to reduce the frequency and severity of metabolic decompensations, but catabolic ketoacidosis and consequent neurological events may still occur due to propionic acid accumulation [22,23]. Previous studies demonstrated that the survival rate after LT in PA and MMA is between 50% and 75% [24,25]; our limited experience is consistent with these findings. 

Cardiac screening following a diagnosis of OAs is important to guide treatment and stratify risk. Kovacevic et al. (2020) demonstrated that, when both systolic and diastolic LV dysfunction were diagnosed in PA patients, the latter developed considerably earlier. Therefore, assessing diastolic LV function during routine cardiac evaluation is useful to determine cardiac involvement at an early stage. This finding may lead to timely cardiac treatment modification, if diastolic LV dysfunction is present with preserved systolic LV function [26]. 

Previous studies have shown that CMP associated with OAs may improve or even remit following correction of the underlying metabolic derangement [27]. In addition to liver transplantation, other valid therapeutic options include high doses of L-carnitine, because of suspected low carnitine levels in skeletal and cardiac muscle [28]. In some cases, despite optimized treatment, cardiac dysfunction may be progressive and require cardiac transplantation. More recently, multiple gene therapy approaches have been explored in relevant animal models [29,30]. However, experience is limited, and there is concern that the operative stress may result in a major episode of metabolic decompensation and complicate the post-operative course. 

## 5. Conclusions

The phenotypic presentation and disease onset of organic acidurias may be extremely variable and complex; affected individuals may present with life-threatening acute metabolic crises and acute multi-organ failure, even during the newborn period. Among late-disease complications, dilated cardiomyopathy represents a common finding in some etiologies of OAs, but the underlying mechanisms are not clear, due to the rarity and complexity of these disorders. Moreover, cardiac involvement seems largely independent from metabolic status. The presence of cardiomyopathy in OAs is associated with unfavorable prognosis; an early onset of symptoms, a diagnosis of PA and specific subtypes of MMA are indicators of poor prognosis. Recent diagnostic and therapeutic strategies have improved the outlook of individuals with OAs. However, the long-term prevention of cardiac complications remains challenging. 

## Figures and Tables

**Figure 1 diagnostics-13-03674-f001:**
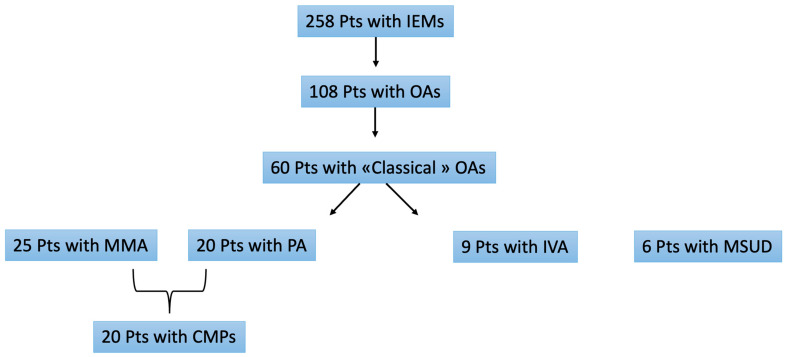
Prevalence of OAS in our study cohort.

**Figure 2 diagnostics-13-03674-f002:**
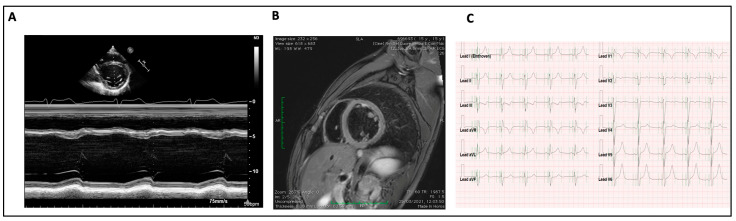
Dilated cardiomyopathy in propionic acidemia in 12-year-old patient. (**A**) Echocardiography, M-Mode short axis: left ventricle is dilated, with low ejection fraction. (**B**) Cardiac Magnetic resonance: dilated left ventricle without fibrosis. (**C**) Electrocardiography: sinus rhythm, HR of 61 beats per minute. Qtc interval prolongation (480 msec).

**Figure 3 diagnostics-13-03674-f003:**
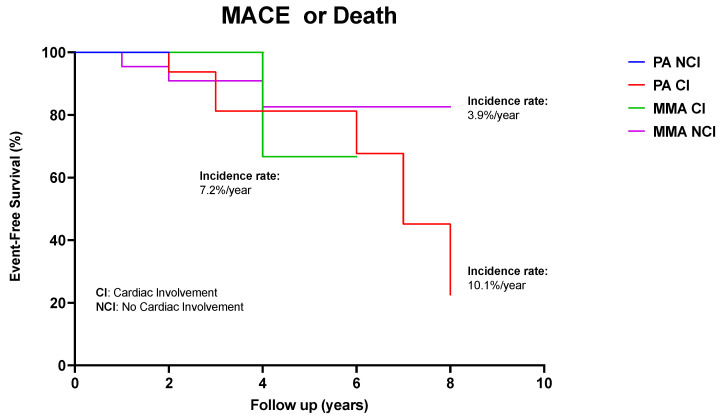
Kaplan–Meier survival analysis showing impaired survival (due to all-cause mortality) in patients with cardiac involvement and without CI involvement. Time zero is defined as time at diagnosis of metabolic disorder.

**Table 1 diagnostics-13-03674-t001:** Baseline and follow-up assessment of patients diagnosed with organic acidemias.

Family History	Pattern of transmission (AR) Recurrence of distinctive features Myopathy Mental retardation/epilepsy—preco cious death/stillbirth Cardiovascular diseases
Clinical examination	Auxological parameters (weight, length/stature, dysmorphias, cranial circumference) Vital parameters (blood pressure, heart rate, oxygen saturation) Assessment of neurological status and cognitive development
Cardiologic evaluation	Clinical examination 12-lead ECG Transthoracic echocardiography Holter ECG CMR
Laboratory investigation	Blood: ↑ Propionyl carnitine ↑Glycine level Urine: ↑ amino acids, organic acids Acylcarnitine profile

**Table 2 diagnostics-13-03674-t002:** Clinical features, diagnosis and outcome in enrolled patients.

	Overall	PA	MMA	IVA	MSUD
Number of patients	60	20 (33.3%)	25 (41.6%)	9 (15%)	6 (10%)
Male patients	32 (53%)	8 (40%)	14 (56%)	8 (88.9%)	2 (33.4%)
Mean age of diagnosis of OAs	<1	<1	<1	<1	<1
Family history	7 (11.7%)	4 (20%)	1 (4%)	2(22.2%)	0
NBS diagnosis	17 (28.3%)	2 (10%)	8 (32%)	4 (44.5%)	3 (50%)
Clinical diagnosis	36 (60%)	14 (70%)	16 (64%)	3 (33.4%)	3 (50%)
Molecular analysis	42(70)	12(60%)	23 (92%)	7(77.8%)	0
Neurological involvement	28 (46.7%)	12 (60%)	8 (32%)	5 (55.6%)	3 (50%)
Ocular involvement	5 (8.3%)	2 (10%)	2 (8%)	0	1 (16.7%)
Renal involvement	5 (8.3%)	1 (5%)	3 (12%)	0	1 (16.7%)
Failure to thrive	4 (6.7%)	2 (10%)	1 (4%)	1 (11.2%)	0
Hematological involvement	5 (8.3%)	2 (10%)	3 (12%)	0	0
Gastrointestinal	7 (11.7%)	2 (10%)	3 (12%)	1 (11.2%)	1 (16.7%)
Skeletal involvement	6 (10%)	3 (15%)	1 (4%)	0	2 (33.4%)
Cardiac involvement	23 (38%)	17 (85%)	6(24%)	0	0
CMP in CI	20 (33.3%)	17(85%)	3 (12%)	0	0
Mean age of CMP onset	10.3	10.2	11	0	0
Average EF% at first valuation	53.3	52.76	56.7	0	0
Average EF% at last valuation	46.9	46.4	50	0	0
Average QTc (Sec)	465.5	465.3	466.7	0	0
MACEs with CI (%/year)	9%	10%	7.2%	0	0
Death for HF	2 (3.3%)	2 (10%)	0	0	0
Hospitalization for HF	4 (6.6%)	4 (20%)	0	0	0
Metabolic Crisis	4(6.6%)	3 (15%)	1 (4%)	0	0
Liver transplantation	7 (11.7%)	5 (25%)	2(8%)	0	0
Mortality	5 (8,3%)	4 (20%)	1 (4%)	0	0
Cardiac cause	2 (10%)	2 (10%)	0	0	0
Non-cardiac cause	3 (15%)	2 (10%)	1 (4%)	0	0

Abbreviations: NBS—new born screening; CMP—cardiomyopathy; CI—cardiac involvement; MACEs—major cardiovascular events.

## Data Availability

Data are contained within the article.

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
