# Peer review of "Cardiac Involvement in Classical Organic Acidurias: Clinical Profile and Outcome in a Pediatric Cohort"

_diagnostics, 2023, doi:10.3390/diagnostics13243674_

Round 1

Reviewer 1 Report

Comments and Suggestions for Authors

This study indicates that dilated cardiomyopathy represents a common finding in some groups of organic acidurias and is associated with unfavorable prognosis, hence is clinically significant and interesting for clinicians. 

The “Methods” were described too simply, which should be described in more detail, especially for genetic tests.

Comments on the Quality of English Language

There are too many paragraphs, of which most may be incorporated.

There are many grammatical/spelling errors to be corrected.

Author Response

Thank you very much for taking the time to review this manuscript. I expanded the methods, especially in the field of genetics; additions are inserted in red in the re-submitted files.

  • Isolated MMA can be caused by defects in four different genes including MMUT, MMAA (cblA), MMAB (cblB), MMADHC (cblD-MMA). cblA and cbLB variants are responsive to cobalamin supplementation. Pag 3. Line 94
  • The subtype mut0 is unresponsive to hydroxocobalamin supplementation in the propionate incorporation assay, while mut− subtype is characterized by increased propionate incorporation after addiction of hydroxocobalamin to the cell culture medium. Pag 3. Line 99
  • There is no evidence of genotype-phenotype correlation in PA, so the assessment of gene defect (in PCCA or PCCB) is not decisive in driving clinical management or prognosis. Pag 3. Line 104
  • The most frequent missense mutation was p.A282V on the IVD gene. Diagnosis of IVA was confirmed by various direct and indirect functional tests to assess IVD activity, in combination with molecular genetic analyses Pag 3. Line 109
  • Patients are usually homozygous or compound heterozygous for mutations in the same gene; there is no evidence of heterozygotes for mutations in two different genes. Pag 3. Line 112
  • The quality of English Language has been improved.

 We appreciate all the comments from the reviewers.

Thank you for reviewing our manuscript

Sincerely,

Silvia Passantino

Reviewer 2 Report

Comments and Suggestions for Authors

Passantino et al described cardiac involvement in organic acidemias in a pediatric cohort of 60 patients. The study is interesting, but in my opinion there is an important bias: very different diseases are analysed, but the results are expressed in an aggregate way, while they should be distinguished by pathology.

Minor comments:

Introduction:

- line 33: the incidence ranges.... based on what? geographical origin, type of pathology?

-lines 48-50: please, clarify the pathogenesis well

lines 51-54: please indicate the available therapies

Author Response

Thank you very much for taking the time to review this manuscript. I expanded the methods, especially in the field of genetics; additions are inserted in red in the re-submitted files.

  1. The incidence of OAs varies in accordance with the study population based on age, referral patterns and ethnicity.  The estimated incidence ranges based between 1:10.000 and 1:1.000.000 live births.  Pag 2. Line 45 pag
  2.  The pathophysiology of cardiac complications has not been completely elucidated, but is thought to reside in an interplay of mitochondrial dysfunction (bioenergetic deficiency and oxidative damage), miRNA deregulation, histone deacetylase (HDAC) inhibition by propionate. Pag. 2 Line 63-64.

  3. The first objective of the pharmacologic therapy, especially in patients with PA and MMA is to prevent secondary carnitine deficiency, with carnitine supplementation. The maintenance treatment includes L-carnitine supplementation, low protein diet and precursor-free synthetic amino acids mixtures, minerals and micronutrients. Despite the use of effective therapy with a low protein diet and carnitine the overall outcome results poor. Pag 2 Line 66

  • The quality of English Language has been improved.

 We appreciate all the comments from the reviewers.

Thank you for reviewing our manuscript

Sincerely,

Silvia Passantino

Round 2

Reviewer 2 Report

Comments and Suggestions for Authors

The paper has been improved